# Minitron-SSM: Efficient Hybrid Language Model Compression through Group-Aware SSM Pruning

**Ali Taghibakhshi,**\* **Sharath Turuvekere Sreenivas,**\* **Saurav Muralidharan**\*
**Marcin Chochowski,**\* **Yashaswi Karnati,**\* **Raviraj Joshi, Ameya Sunil Mahabaleshwarkar**
**Zijia Chen, Yoshi Suhara, Oluwatobi Olabiyi, Daniel Korzekwa**
**Mostofa Patwary, Mohammad Shoeybi, Jan Kautz, Bryan Catanzaro**
**Ashwath Aithal, Nima Tajbakhsh, Pavlo Molchanov**

**NVIDIA**

{ataghibakhsh, sharatht, sauravm, mchochowski, ykarnati, ravirajj,
ameyasunilm, zijiac, ysuhara, oolabiyi, dkorzekwa, mpatwary,
mshoeybi, jkautz, bcatanzaro, aaithal, ntajbakhsh, pmolchanov}@nvidia.com

## Abstract

Hybrid language models that combine Attention and State Space Models (SSMs) have been shown to achieve state-of-the-art accuracy and runtime performance. Recent work has also demonstrated that applying pruning and distillation to Attention-only models yields smaller, more accurate models at a fraction of the training cost. In this work, we explore the effectiveness of compressing Hybrid architectures. To this end, we introduce a novel group-aware pruning method for Mamba layers that preserves the structural integrity of SSM blocks and their sequence modeling capabilities. We combine this method with FFN, embedding dimension, and layer pruning, along with knowledge distillation-based retraining to obtain a unified compression recipe for hybrid models. Using this recipe, we compress the Nemotron-H 8B Hybrid model down to 4B parameters with up to $40\times$ fewer training tokens compared to similarly-sized models. The resulting model surpasses the accuracy of similarly-sized models while achieving $\sim 2\times$ faster inference throughput, significantly advancing the Pareto frontier.

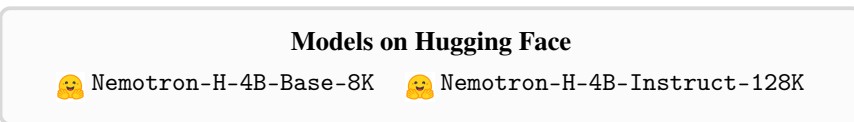

**Models on Hugging Face**

🤗 Nemotron-H-4B-Base-8K    🤗 Nemotron-H-4B-Instruct-128K

## 1 Introduction

Recent advances in language modeling have led to the development of hybrid architectures that combine Transformer layers [25] with State Space Models (SSMs) [11, 7]. These hybrid models leverage the complementary strengths of both approaches: Transformers excel at capturing global dependencies through self-attention mechanisms, while SSMs provide efficient sequence processing with $O(N)$ scaling during training and $O(1)$ cache size during inference. Mamba [11, 7] in particular is a popular SSM designed for efficient sequence modeling with linear-time complexity and support for long contexts and is often the preferred choice for non-attention layers in hybrid architectures.

---

\*Equal contribution.

39th Conference on Neural Information Processing Systems (NeurIPS 2025).

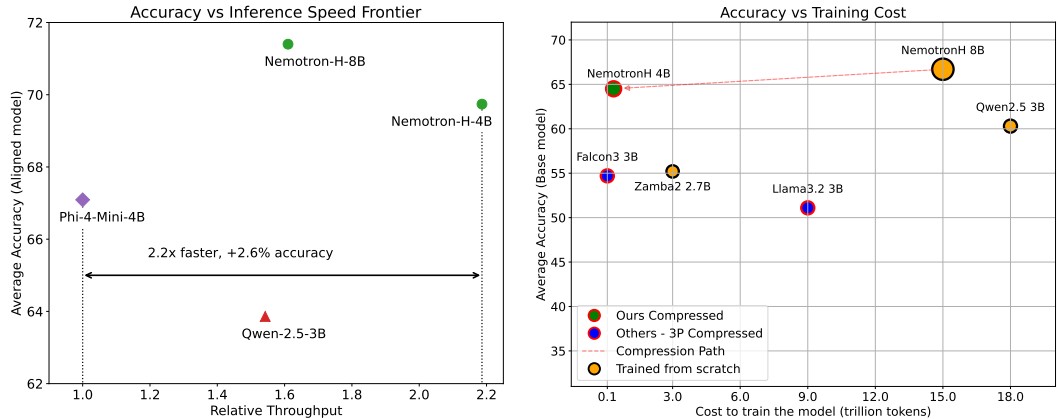

Figure 1: Comparison of NEMOTRON-H 4B model accuracy w.r.t. inference throughput (left), and training budget for the base model (right) to similarly-sized community models. Inference throughput is measured at an input and output sequence length of 65536 and 1024, respectively.

However, despite their improved efficiency, many hybrid LLMs remain incredibly large, often spanning billions of parameters - this motivates the need for efficiently creating smaller hybrid models suitable for deployment in resource-constrained environments.

Model pruning—the removal of redundant parameters while preserving accuracy—has recently emerged as a promising approach for compressing LLMs. In particular, methods that combine structured pruning (i.e., pruning of entire parameter blocks such as neurons, attention heads, etc.) with knowledge distillation [13] have proven effective at simultaneously reducing model memory footprint while improving runtime performance and accuracy [20].

While pruning techniques have been extensively studied for Transformer architectures [20, 4, 24], their application to hybrid models remains significantly underexplored. Indeed, the optimization of SSMs is an active area of research, with various methods being explored, including post-training quantization [6].

Regarding pruning specifically, some early work on Mamba and SSMs includes Mamba-Shredder [19], which removes the entire state space module from the Mamba layers, leaving only linear projections and a convolution layer. In a concurrent study, Ghattas et al. [9] propose a method for pruning Mamba architectures by focusing on three aspects: state space dimension reduction, Mamba head dimension pruning, and Mamba head merging.

To the best of our knowledge, no existing work on SSM/Mamba pruning presents a holistic compression strategy that simultaneously combines various aspects of SSM pruning with the pruning of other network components such as FFN neurons, embedding channels, and network depth; we believe such an approach is essential for obtaining the best combination of runtime performance and model accuracy. In this paper, we introduce a novel group-aware pruning method for Mamba layers that targets both Mamba heads and head channels. We combine this method with FFN, embedding dimension, and layer pruning to obtain a unified compression recipe for hybrid models that maximizes their accuracy and runtime performance.

This paper makes the following key contributions:

- Introduces a group-aware pruning method for Mamba layers that preserves SSM block structure and sequence modeling capabilities.
- Presents a novel *hybrid pruning recipe* that effectively combines Mamba pruning with the pruning of other network components such as FFN neurons, embedding channels and layers.
- Presents findings on the sensitivity of Mamba block components to pruning, along with accuracy-throughput trade-offs when combined with pruning of other network components.
- Utilizes the proposed hybrid pruning recipe to compress the Nemotron-H 8B model to 4B parameters through pruning and knowledge distillation. As shown in Figure 1, the resulting

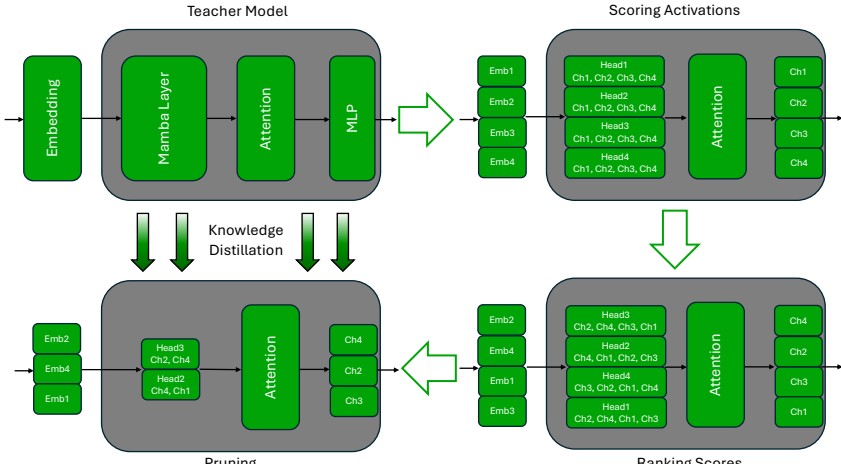

Figure 2: Overview of pruning and distillation for hybrid architectures. Starting from a pretrained LLM, we first evaluate the importance of Mamba heads and channels, FFN neurons, and embedding channels. We then rank them, trim the least important neurons, and distill the knowledge from the original LLM to the pruned model. Attention layers are not pruned since they amount to only 8% of the total number of layers.

model requires up to $40\times$ fewer training tokens compared to others in the same size range. It also achieves state-of-the-art accuracy on benchmarks, and a $\sim 2\times$ speedup in inference throughput compared to similarly-sized models, significantly pushing the Pareto frontier.

## 2  Pruning & Distillation Methodology

We start the pruning procedure by computing the importance or sensitivity of each network component; namely, Mamba heads and head channels, FFN neurons, embedding channels, and layers. To keep this phase lightweight, we adopt a purely activation-based strategy (requiring only forward propagation passes) for computing importance scores, similar to Minitron [20]. Once scores are computed, we sort the corresponding network components in decreasing order of importance while following any additional implementation constraints (discussed in more detail later in Section 2.1). We then prune away the network components with the lowest scores. Finally, the pruned model is distilled using the teacher model to obtain the final pruned model. The full procedure is illustrated in Figure 2.

### 2.1  Mamba Pruning

**Background - State Space Models (SSMs).** SSMs are a class of sequence models that process inputs through hidden states evolving over time [7]. The general form of an SSM is given by:

$$h_t = Ah_{t-1} + Bx_t \quad (1) \qquad\qquad y_t = C^\top h_t + Dx_t \quad (2)$$

Here, $h_t$ represents the hidden state, $x_t$ the input, $y_t$ the output, and $A$, $B$, $C$, and $D$ are parameter matrices. The above equations describe linear time-invariant (LTI) SSMs, where the parameters remain constant across timesteps. The Mamba architecture [7] introduced a selective SSM variant with time-varying parameters:

$$h_t = A_t h_{t-1} + B_t x_t \quad (3) \qquad\qquad y_t = C_t^\top h_t + D_t x_t \quad (4)$$

This selective mechanism allows the model to adapt dynamically to the input sequence, improving performance on complex tasks. Mamba2 [7] builds upon the selective SSM framework and introduces several enhancements to improve efficiency and scalability. It leverages the Structured State Space Duality (SSD), which connects SSMs and attention mechanisms through semi-separable matrix representations. This duality enables Mamba2 to combine the linear efficiency of SSMs with hardware-friendly quadratic computations typical of attention models.

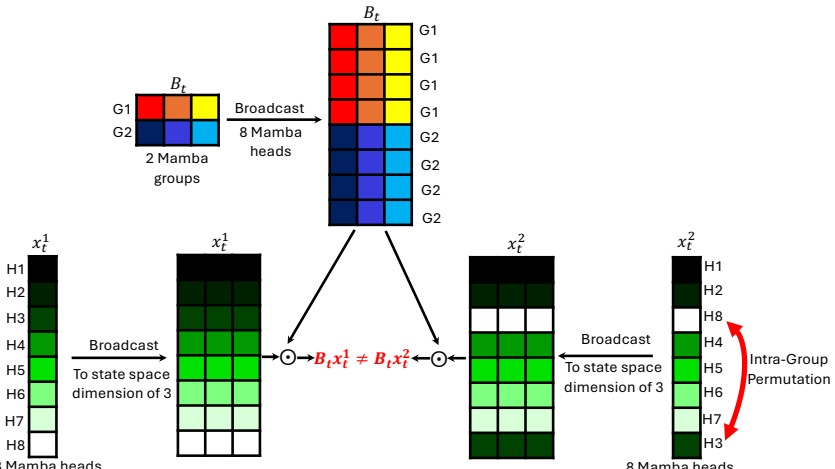

Figure 3: Mamba group structure visualization showing broadcasting and original $B_t x_t$ computation. Colors represent distinct entries. The Figure illustrates how only within-group head permutations can preserve SSM semantics. As a counter example, if H3 and H8 were to be swapped, the resulting $B_t x_t$ would NOT be any permutation of the original (no permutation) $B_t x_t$.

To better understand importance estimation and pruning of Mamba layers, we first dive into the forward pass of a Mamba layer. The Mamba layer processes input through five distinct projection matrices $W_z, W_x, W_B, W_C$, and $W_{d_t}$, following layer normalization. Let us denote the input to the Mamba layer as $X$. Then, these projections generate intermediate matrices [2] $z = W_z(\text{LN}(X))$, $x = W_x(\text{LN}(X))$, $B = W_B(\text{LN}(X))$, $C = W_C(\text{LN}(X))$, and $d_t = W_{d_t}(\text{LN}(X))$, which have the following dimensions, respectively; $W_z, W_x \in \mathbb{R}^{d_e \times (m_h \times m_d)}$; $W_B, W_C \in \mathbb{R}^{d_e \times (g \times d_s)}$, and $W_{d_t} \in \mathbb{R}^{d_e \times m_h}$. LN denotes layer normalization, $d_e$ is the model embedding dimension (a.k.a hidden dimension), $g$ is the number of Mamba groups, $d_s$ is the SSM state dimension, $m_h$ is the number of Mamba heads, and $m_d$ is the Mamba head channels. The matrices $x, B$, and $C$ undergo causal convolution, as in $\hat{x} = \text{conv1d}(x)$, $\hat{B} = \text{conv1d}(B)$, $\hat{C} = \text{conv1d}(C)$. Consequently, the obtained matrices participate in the selective state space model (SSM) updates, $\tilde{y} = \text{SSM}(\hat{x}, \hat{B}, \hat{C}, A, D, d_t)$. Here, $A, D \in \mathbb{R}^{m_h}$ are SSM learnable parameters corresponding to state transition and direct feed through, respectively (see Equations 3 and 4). The SSM output is fed into a gated normalization layer, which is then followed by output projection, $y = W_O(\text{RMSNorm}(\tilde{y}, z))$, where $W_O \in \mathbb{R}^{(m_h \times m_d) \times d_e}$ is the output projection.

**Group-Aware Head Permutation Constraints.** Pruning requires scoring, sorting, and trimming neurons or heads of each layer, as shown in Figure 2. The FFN and embedding activations are permutation equivariant, i.e. for a permutation operator $\mathcal{P}$, FFN or embedding layer $L$, and activation $\mathcal{A}$, and input $X$ we have $L(X) = \mathcal{A} \implies \mathcal{P}(L)(X) = \mathcal{P}(\mathcal{A})$. However, Mamba layers and activations are not permutation equivariant. As shown in Figure 3, the $B_t x_t$ operation from Eq. 3 involves reshaping $B$ into $B \in \mathbb{R}^{g \times d_s}$, and broadcasting it across $x \in \mathbb{R}^{(m_h \times m_d)}$. This broadcasting creates group-specific interaction patterns that constrain our pruning approach. As a result, permuting heads across groups would alter the $B_t x_t$ broadcast pattern, violating Eq. 3's group-wise computation as shown by $B_t x_t \neq (B \mathcal{P}(x_t)) \mathcal{P}^T$. Therefore, when sorting Mamba heads using activation scores, we must preserve Mamba's group structure. Let $\mathcal{G}_g \subset \{1, ..., m_h\}$ denote the set of heads belonging to group $g$. Any permutation $\mathcal{P}$ of heads must satisfy $\mathcal{P}(h) \in \mathcal{G}_g \quad \forall h \in \mathcal{G}_g$. In other words, Mamba heads and activations are permutation equivariant only for the aforementioned permutation operators.

**Head Channel Consistency.** A similar constraint for permuting Mamba head channels applies. For head channel pruning, we maintain consistency across all heads through shared ranking. The state tensor $h \in \mathbb{R}^{m_h \times m_d \times d_s}$ requires channel-wise permutations $\mathcal{P}_d$ to satisfy $\mathcal{P}_d(h_{i,j,k}) = \mathcal{P}_d(h_{i',j,k}) \quad \forall i, i' \in \{1, ..., m_h\}$, meaning each channel index $k$ is either preserved or pruned uniformly across all heads.

---

[2]We factor out the sequence length and batch size to simplify our description; the analysis remains valid without them.

**Scoring and Ranking Methodology.** The Mamba head and head channel ranking follows a nested scoring procedure:

1. **Head Channel Scoring**: For each head channel $d \in \{1, ..., m_d\}$, we compute aggregate importance scores $s_d = \|\sum_{B,L} s_{:,d}\|_2$ where $s = LN(X)(W_x)^T$, and the aggregation is over $L$, the sequence length, and $B$, the batch size. Aggregation metric used along $L$ and $B$ dimensions are mean and $L_2$, respectively, following Minitron [20]. $s \in \mathbb{R}^{(m_h \times m_d)}$ contains raw activation scores, and $\mathbf{s}_{:,d}$ denotes the $d$-th column across all heads. We then select the top-$k_d$ channels, $\mathcal{D}_{\text{top}} = \underset{d \in \{1, ..., m_d\}}{\text{topk}} (s_d, k = k_d)$.

2. **Head Scoring**: Using the pruned channels $\mathcal{D}_{\text{top}}$, compute head importance scores, $f_h = \|\mathbf{s}_{h, \mathcal{D}_{\text{top}}}\|_2 \quad \forall h \in \{1, ..., m_h\}$.

3. **Group-Constrained Ranking**: Within each Mamba group $\mathcal{G}_g$, sort heads by their scores, $\mathcal{R}_g = \underset{h \in \mathcal{G}_g}{\text{argsort}}(f_h)$.

The final head ranking $\mathcal{R}$ is the concatenation of group-wise rankings, $\mathcal{R} = \bigoplus_{g=1}^{G} \mathcal{R}_g[1 : k_g]$, where $k_g$ is the target head count per group and $\bigoplus$ denotes ordered concatenation.

The following algorithm provides a concise walkthrough of Mamba head and head channel ranking:

---

**Require:** Activation scores $\mathbf{s} \in \mathbb{R}^{m_h \times m_d}$, target channels $k_d$, target heads per group $\{k_g\}_{g=1}^{G}$
**Ensure:** Head ranking $\mathcal{R}$, channel ranking $\mathcal{D}_{\text{top}}$
1: Compute channel scores: $s_d \leftarrow \|\mathbf{s}_{:,d}\|_2 \ \forall d$
2: $\mathcal{D}_{\text{top}} \leftarrow$ top-$k_d$ indices of $\{s_d\}$
3: Compute head scores: $f_h \leftarrow \|\mathbf{s}_{h, \mathcal{D}_{\text{top}}}\|_2 \ \forall h$
4: **for** $g \leftarrow 1$ **to** $G$ **do**
5: $\quad \mathcal{R}_g \leftarrow$ argsort-descending($\{f_h \mid h \in \mathcal{G}_g\}$)
6: $\quad \mathcal{R}_g^{\text{sel}} \leftarrow$ first $k_g$ elements of $\mathcal{R}_g$
7: **end for**
8: $\mathcal{R} \leftarrow \bigoplus_{g=1}^{G} \mathcal{R}_g^{\text{sel}}$

---

After obtaining the Mamba heads and head channel neurons to keep, we trim the corresponding matrices, $W \leftarrow W[\mathcal{R}]$, for $W \in \{W_x, W_z, W_O, W_A, W_D, W_{d_t}, \text{conv1d}\}$.

## 2.2 Pruning FFN, Embedding, and Depth

Similar to the Minitron approach for structured pruning of Transformers [20], we examine the activations produced by the FFN and LayerNorm layers to determine which neurons and embedding channels contribute least to the model's performance. Formally, for the $i$-th neuron in a feed-forward layer, we compute its importance score as: $F_{\text{neuron}}^{(i)} = \sum_{B,L} X(W_1^i)^T$, where $W_1^i$ refers to the $i$-th row of the weight matrix $W_1$ in the first linear projection of the FFN, $X$ is the input to the FFN layer, and $\sum_{B,L}$ denotes aggregation along the batch and sequence dimensions. Similarly, for the $i$-th embedding channel, we compute: $F_{\text{emb}}^{(i)} = \sum_{B,L} LN(X)_i$ where $LN(X)_i$ represents the $i$-th dimension of the layer-normalized input. The embedding channel scores are computed across all layers that utilize the embedding channel, including FFN, Mamba and Attention projection layers, and LayerNorm components. We use mean and L2-norm aggregation along $B$ and $L$ dimensions, respectively. After computing these scores, we sort them in descending order and keep the top-k neurons and embedding channels based on the target compression ratio, pruning those with the lowest importance scores.

**Depth Pruning.** We analyze layer importance using Kullback-Leibler divergence (KLD) between logits from a model with a specific layer removed and the full model. This importance estimation is averaged over a small random subset of 256 samples to account for sample variability. The layer(s) with lowest importance scores are subsequently removed. We provide a detailed analysis of importance distribution for the Nemotron-H 8B model in Section A.2.

**FLAP Importance.** FLAP [2] is a retraining-free structured pruning technique designed to measure the recoverability of a model's output feature map upon removing specific columns from weight

matrices. While we explored FLAP as an alternative importance metric, we observed that it didn't offer any clear advantage over activation-based importance (see Section A.1 for more details).

## 2.3 Architecture Search

Our compression strategy explores multiple axes within the 4B parameter budget through combinatorial pruning. Our search space includes depth reduction (removing 4-26 layers from the original 52-layer architecture) combined with width pruning of embedding channels (3072-4096), FFN dimension (9984-21504), Mamba heads (64-128), and Mamba head channels (32-64). This multi-axis search space includes 125 candidate architectures meeting the parameter constraints.

Our search procedure follows these steps: (1) compute the zero-shot validation loss for all candidates on 1024 calibration samples, (2) select the top K architectures (22 in this paper) with the best loss values and perform lightweight knowledge distillation (KD) on them with 3.8B tokens, using the original 8B model as the teacher, and (3) select the top architecture candidate from step (2), using throughput and latency measurements for breaking ties, and perform extended knowledge distillation with $\sim 380B$ tokens to obtain the final model (see Table 2). We note that step (2) is critical for getting a reliable ranking of architectural candidates, as also noted in prior work [20].

## 2.4 Accuracy Recovery with Knowledge Distillation (KD)

To recover the accuracy lost due to pruning, the model undergoes continued training. Recent work has demonstrated that distilling knowledge [13] from the original model to the pruned model outperforms conventional fine-tuning [23, 4]; we thus adopt logit-based distillation for continued training, employing forward KL divergence (FKLD) loss exclusively during the accuracy recovery phase, following Minitron [20].

# 3 Experiments and Results

To identify the optimal compression strategy for hybrid models, we conduct several ablation studies evaluating the impact of pruning different components on accuracy and runtime performance. Our experiments, summarized in Section 3.1, reveal key insights and highlight differences from Transformer-only compression [3]. We then describe our main results for the NEMOTRON-H 4B model in Section 3.2.

## 3.1 Analysis and Ablations

**Depth vs Width Pruning.** As shown in Table 1, width-only pruning (#1) significantly outperforms depth-only pruning (#24) at a 50% compression ratio (8B to 4B). Notably, a depth-pruned model with 36 layers (#25), despite having ~1.4× more parameters performs worse than the least accurate width-only pruned 4B candidate (#23, with 64 Mamba heads), demonstrating the critical role of depth in maintaining accuracy as also observed with Transformer-only models.

> **Finding 1:** Width pruning considerably outperforms depth pruning in terms of accuracy.

**Impact on Inference Throughput.** Table 1 shows that depth-only pruning (#24) provides the highest inference speedups. Figure 7 (Appendix) presents the correlation between pruning various network components and performance metrics such as throughput, latency, and LM-loss for a fixed 4B parameter count. We notice from the Figure that pruning Mamba components results in faster models compared to pruning FFN and embedding dimensions. Furthermore, we also compare the effects of pruning Mamba heads to pruning head channels in Figure 8 (Appendix); we observe that the former yields better speed improvements than the latter within a given Mamba layer.

> **Finding 2:** Impact on throughput: Depth pruning > Mamba > FFN and Embedding.

---

[3]Note that due to space restrictions, we provide most of the Tables/Figures related to ablations in the Appendix.

| # | Layers | Emb | FFN | Heads | Head Channel | LM Val Loss | Relative Throughput |
|---|--------|-----|-----|-------|--------------|-------------|---------------------|
| **1** | **52** | **3072** | **12288** | **112** | **64** | **1.380** | **1** |
| 2 | 52 | 3072 | 10752 | 128 | 64 | 1.380 | 0.98 |
| 3 | 52 | 3328 | 9984 | 112 | 64 | 1.384 | 1 |
| 4 | 52 | 3072 | 12288 | 112 | 60 | 1.388 | 1.02 |
| 5 | 52 | 3072 | 12288 | 120 | 56 | 1.388 | 1.01 |
| 6 | 52 | 3072 | 13056 | 112 | 56 | 1.389 | 1.04 |
| 7 | 44 | 3072 | 14592 | 128 | 64 | 1.393 | 1.11 |
| 8 | 44 | 3584 | 10752 | 120 | 64 | 1.394 | 1.12 |
| 9 | 52 | 3072 | 11520 | 112 | 64 | 1.396 | 1.02 |
| 10 | 52 | 3072 | 13056 | 96 | 64 | 1.396 | 1.04 |
| 11 | 52 | 3072 | 13824 | 128 | 48 | 1.396 | 1.03 |
| 12 | 52 | 3072 | 12288 | 104 | 62 | 1.397 | 1.03 |
| 13 | 52 | 3072 | 13056 | 104 | 60 | 1.397 | 1.03 |
| 14 | 52 | 3072 | 13056 | 96 | 62 | 1.397 | 1.04 |
| 15 | 52 | 3072 | 14592 | 96 | 56 | 1.398 | 1.05 |
| 16 | 48 | 3072 | 12288 | 128 | 64 | 1.398 | 1.08 |
| 17 | 48 | 3328 | 9984 | 128 | 64 | 1.399 | 1.07 |
| 18 | 52 | 3072 | 13824 | 96 | 58 | 1.401 | 1.05 |
| 19 | 52 | 3072 | 11520 | 128 | 56 | 1.402 | 1.01 |
| 20 | 44 | 3328 | 11648 | 128 | 64 | 1.402 | 1.12 |
| 21 | 48 | 3072 | 13824 | 112 | 64 | 1.403 | 1.09 |
| 22 | 48 | 3328 | 11648 | 112 | 64 | 1.403 | 1.08 |
| 23 | 52 | 3072 | 16128 | 64 | 64 | 1.411 | 1.07 |
| 24 | 26 | 4096 | 21504 | 128 | 64 | 1.533 | 1.31 |
| 25* | 36 | 4096 | 21504 | 128 | 64 | 1.430 | 1.12 |
| 8B parent | 52 | 4096 | 21504 | 128 | 64 | - | 0.74 |

Table 1: Model configurations with their corresponding LM validation loss after lightweight KD (sorted in increasing order), and relative inference throughput. Highlighted row shows the best (lowest) loss. All models have $\sim 4B$ parameters, except entries marked with *, which have more.

**Impact on Accuracy.** Table 1 shows that model depth (#24) is most sensitive to accuracy, followed by Mamba heads (#23), while FFN and embedding dimensions have less impact. Further ablations isolating the pruning of Mamba heads and head channels show that pruning head channels leads to a greater accuracy loss (Figure 8). Given depth pruning's effect on inference speed, we explore a combined pruning strategy, starting with depth-only pruning followed by distillation to assess its limits. As shown in Figure 5, we observe significant accuracy drops on math and coding benchmarks below 44 layers. We then apply width pruning to both the 44- and 48-layer variants to produce corresponding ∼4B-sized models. However, we notice that the best depth-width pruned candidate (#7, 44 layers) still under-performs the width-only model (#1).

> **Finding 3:** Combining depth and width pruning mitigates the degradation seen with depth-only pruning, but still under-performs width-only pruning.

**Mamba Layer Scoring.** As described in Section 2.1, we choose the activations obtained from the $W_x$ matrix for scoring the Mamba heads and head channels. We can alternatively get Mamba scores by considering the activations obtained from $W_z$ and $W_O$ matrices. Table 7 shows the effect of selecting the Mamba activations from different parts of the Mamba layer. For different configurations, we notice that scoring the activations from $W_x$ output often results in the best LM-loss.

> **Finding 4:** Scoring Mamba heads using activations from the $W_x$ matrix yields the best LM-loss.

**Effect of Parameter Choice on Performance Metrics.** As described in Section 2.3, we obtain 125 checkpoints that fit the 4B parameter size constraint with different FFN channels, embedding dimensions, $m_h$ (Mamba heads), and $m_d$ (Mamba head channels). For each checkpoint, we evaluate the LM-loss, time to first token, and throughput. To analyze the relationships between model parameters and performance metrics, we compute correlations and visualize them in Figure 7. Additionally, since all 125 models have the same total parameter count (4B), the model parameters exhibit negative correlations with one another. Figure 7 shows that in 4B models derived from Nemotron-H 8B, Mamba components positively correlate with latency and negatively with throughput and LM-loss, indicating that pruning them improves inference speed and slightly degrades accuracy. In contrast, pruning embedding and FFN dimensions improves accuracy (lower LM-loss) but leads to slower models with increased latency and reduced throughput.

> **Finding 5:** Pruning Mamba components improves inference speed but slightly degrades accuracy, while pruning embedding and FFN dimensions improves accuracy but slows down models.

**Closer Look at Mamba Pruning.** We analyze the sensitivity of pruning two axes in the Mamba layer—Mamba heads ($m_h$) and Mamba head channels ($m_d$)—to various metrics, including accuracy, latency, and throughput. In this study, each axis was pruned in isolation while keeping the rest of the network unchanged, preserving the architecture of the Nemotron-H 8B model. The objective was to determine which axis is more favorable for optimization. As shown in Figure 8, pruning Mamba heads ($m_h$) consistently outperforms pruning Mamba head channels ($m_d$) across all metrics. Specifically, reducing $m_h$ consistently yields lower LM-loss, reduced latency, and higher throughput, making Mamba heads a particularly impactful and practical target for pruning.

> **Finding 6:** Pruning Mamba heads ($m_h$) consistently improves both accuracy and speed compared to Mamba head channels ($m_d$).

**Transformer vs. Hybrid Pruning.** The above findings highlight the importance of choosing the right pruning axes in hybrid models to balance accuracy and efficiency. Unlike Transformer-only models where pruning attention heads is less common [20], hybrid architectures can tolerate some head pruning, as seen with candidates #1 and #2 in Table 1. This tolerance may stem from Mamba layers having significantly more heads (128) than self-attention layers (32).

> **Finding 7:** Head pruning is better tolerated in Hybrid architectures compared to Transformer-only.

### 3.2 Obtaining the Best Compressed Hybrid Model

For our final model, we select the candidate with the lowest LM validation loss in Table 1, which shows our architecture search results after short knowledge distillation. We note that although both candidates #1 and #2 have identical losses, candidate #1 is chosen for extended KD with 380B tokens due to its higher inference throughput, enabled by the reduction in Mamba heads.

**Data and Training Hyperparameters.** We use a random sample from the Phase 3 data mixture employed for training Nemotron-H models [5] for both importance estimation and knowledge distillation (KD). For importance estimation, we use 1024 samples with a sequence length of 8192. For KD, the batch size is 768, with a sequence length of 8192, a cosine decay learning rate schedule (starting at 1.6e-4 and decaying to 8e-4), with a 60-step linear warmup.

**Alignment.** We perform Supervised Fine-tuning with Knowledge Distillation (SFT-KD) [22] using the Nemotron-H 8B aligned model as the teacher, along with Reward-aware Preference Optimization (RPO) [1] and NeMo-Aligner [21]. The NEMOTRON-H 4B base model is fine-tuned using supervision from the top-k (100) logits of the teacher over two rounds of SFT-KD: the first round uses math and coding data, while the second round focuses on instruction-following and general chat data. The instruction-tuned model is then further aligned with two rounds of RPO.

**Long Context Extension.** To extend the context length of the aligned NEMOTRON-H 4B model, we perform SFT using data designed for long-context understanding: this training data is derived by manipulating the general domain chat dataset from the second SFT-KD round during alignment. We concatenate conversation turns and introduce long-range dependencies by placing related turns far apart within the extended context. The context length is varied randomly between 128k and 512k tokens, ensuring the model learns to maintain coherence and understanding across longer sequences, enhancing its ability to process information beyond shorter context windows.

**Evaluation Summary.** Tables 2 to 5 present accuracy comparisons between our compressed 4B hybrid model, other similar-sized community models, and the parent 8B hybrid model. As shown in Tables, our 4B model retains over 96% of the original 8B model's accuracy, including safety scores on Garak [8] and AEGIS [10], while improving throughput by ∼1.4x. Compared to other similarly sized community models, it delivers state-of-the-art accuracy across knowledge, math, coding, commonsense reasoning, and reading comprehension tasks, despite being trained on up to ∼40x fewer tokens. It also achieves ∼2.2x higher throughput and ∼1.8x lower latency than the second-best Phi-4-4B model (Figures 1 and 6). The aligned version further leads in math, coding, instruction following, and tool-use tasks. To assess long-context capabilities, we use the RULER

| Benchmarks (shots) | Llama-Minitron 4B-Depth-Base* | Llama-Minitron 4B-Width-Base* | Llama-3.2 3B-Base* | Falcon-3 3B-Base* | Zamba-2 2.7B-Base | Qwen-2.5 3B-Base | NEMOTRON-H 4B-Base* | Nemotron-H 8B-Base |
|---|---|---|---|---|---|---|---|---|
| ARC Challenge (0) | 46.8 | 49.0 | 46.5 | 47.4 | 51.5 | 47.3 | **54.4** | 60.1 |
| ARC Easy (0) | 75.0 | 76.5 | 72.0 | 72.4 | 79.5 | 72.7 | **81.6** | 83.6 |
| CommonsenseQA (0) | 70.9 | 71.7 | 66.5 | 64.4 | 76.2 | **77.1** | 70.2 | 72.7 |
| GSM8K (8) | 19.4 | 40.4 | 27.1 | 66.5 | 55.0 | **75.2** | 69.6 | 77.9 |
| HellaSwag (0) | 69.5 | 74.0 | 74.1 | 65.3 | 76.6 | 73.6 | **77.0** | 81.2 |
| HumanEval (0, pass@1) | 23.2 | 34.4 | 26.8 | 39.6 | 25.0 | 37.8 | **59.8** | 57.3 |
| HumanEval+ (0, pass@1) | 18.9 | 20.7 | 24.4 | 32.3 | 21.3 | 33.5 | **55.5** | 53.7 |
| MBPP (3, pass@1) | 43.2 | 47.9 | 42.0 | 52.1 | 36.2 | 59.9 | **65.0** | 66.9 |
| MBPP+ (0, pass@1) | 39.2 | 38.4 | 40.7 | 40.7 | 32.8 | 50.0 | **61.1** | 58.7 |
| MMLU (5) | 57.8 | 59.8 | 56.3 | 56.7 | 56.8 | 65.6 | **68.1** | 72.7 |
| OpenbookQA (0) | 42.4 | 44.0 | 41.4 | 39.4 | **46.4** | 42.2 | 44.2 | 47.2 |
| PIQA (0) | 75.9 | 77.5 | 78.0 | 75.5 | **80.4** | 78.8 | 79.4 | 82.2 |
| RACE v.3 (0) | 59.6 | 65.6 | 66.7 | 69.7 | 73.7 | **84.5** | 80.9 | 84.0 |
| Social IQA (0) | 46.4 | 47.8 | 46.8 | 45.1 | **51.8** | 49.8 | 45.1 | 45.8 |
| TruthfulQA MC2 (0) | 37.2 | 41.0 | 39.3 | 45.6 | 45.8 | 49.0 | **49.4** | 49.8 |
| Winogrande (0) | 68.8 | 70.4 | 69.5 | 65.0 | **74.3** | 68.4 | 71.3 | 76.3 |
| Average | 49.6 | 53.1 | 51.1 | 54.7 | 55.2 | 60.3 | **64.5** | 66.7 |
| Tokens | 0.1T | 0.1T | 9T | 0.1T | 3T | 18T | 0.38T | 15T |

Table 2: Accuracy comparison of our NEMOTRON-H 4B with other similarly sized base community models. * indicates compressed models.

| Benchmarks (shots) | Phi-4-Mini 4B-Instruct-128k | Qwen-2.5 3B-Instruct-32k | Llama-3.2 3B-Instruct-128k* | Falcon-3 3B-Instruct-32k* | Zamba-2 2.7B-Instruct-4k | NEMOTRON-H 4B-Instruct-128k* | Nemotron-H 8B-Instruct-128k |
|---|---|---|---|---|---|---|---|
| MMLU (0, generative) | 61.88 | 63.25 | 57.36 | 54.27 | 55.32 | **66.96** | 68.7 |
| GSM8K (0) | 87.71 | 83.32 | 78.47 | 77.86 | 66.26 | **88.93** | 90.4 |
| MATH-500 (0) | 70.8 | 65.6 | 48.2 | 48.80 | 29.40 | **76.4** | 77.6 |
| HumanEval (0, pass@1) | 73.17 | 75.0 | 55.49 | 46.34 | 37.20 | **76.2** | 79.3 |
| HumanEval+ (0, pass@1) | 64.63 | 70.12 | 51.83 | 43.29 | 32.93 | **70.85** | 74.4 |
| MBPP (0, pass@1) | 67.46 | 67.72 | 65.61 | 61.37 | 46.30 | **78.6** | 81 |
| MBPP+ (0, pass@1) | 60.31 | 58.47 | 55.29 | 55.03 | 38.62 | **68.25** | 67.7 |
| IFEval Strict (0) | 74.78 | 64.06 | 74.51 | 68.49 | 46.99 | **76.24** | 78.6 |
| MT-Bench (0) | 7.86 | 7.68 | 7.09 | 7.10 | 7.02 | **7.90** | 7.90 |
| BFCL v2 Live (0) | 61.64 | 59.08 | 49.58 | 52.80 | 39.70 | **65.88** | 62.6 |

Table 3: Accuracy comparison for instruction-tuned models. For IFEval, we report the average of prompt strict and instruction strict categories. For BFCL v2, we report live overall accuracy. For MT-Bench, we use GPT-4-Turbo as the judge. * indicates compressed models.

| Context Length | Phi-4-Mini 4B-Instruct-128k | Qwen-2.5 3B-Instruct-32k | Llama-3.2 3B-Instruct-128k* | NEMOTRON-H 4B-Instruct-128k* | Nemotron-H 8B-Instruct-128k |
|---|---|---|---|---|---|
| 16,384 | 34.39 | 83.64 | 77.92 | **86.28** | 91.5 |
| 32,768 | 32.90 | 79.21 | 72.71 | **82.27** | 89.8 |
| 65,536 | 35.01 | 63.80 | 66.42 | **75.95** | 87.6 |
| 131,072 | 20.07 | 23.61 | 59.26 | **63.57** | 81.7 |

Table 4: Average RULER benchmark scores up to 128k context length for aligned NEMOTRON-H 4B and other instruction-tuned models in a similar size range. * indicates compressed models.

benchmark [15]. As shown in Table 4, our model demonstrates strong performance and achieves the highest scores at context lengths up to 128k tokens. Finally, Figure 6 (Appendix) compares latency and throughput across four models: Phi-4-Mini-4B, Qwen-2.5-3B, Nemotron-H 8B, and **NEMOTRON-H 4B (ours)**. Our model achieves the best performance on both axes—delivering the fastest time-to-first-token and highest throughput—effectively advancing the latency-throughput Pareto frontier. In summary, our compression approach successfully produces a model with state-of-the-art accuracy while significantly improving inference speed and reducing training costs.

## 4 Related Work

**Structured Weight Pruning.** Weight pruning is a powerful and well-known technique for reducing model size [20, 26, 14]. In particular, *structured pruning* removes blocks of nonzero elements at once from model weights, making it easier to realize actual hardware speedups; examples of structured pruning techniques include neuron, attention head, convolutional filter, and depth pruning [20, 17, 12, 27, 3, 18, 28, 16]. Recent work [20] has demonstrated that knowledge distillation [13] can be an effective alternative to traditional fine-tuning for post-pruning accuracy recovery.

**Hybrid Model Pruning.** Recent work on SSM pruning has focused on Mamba architectures: Mamba-Shredder [19] proposes removing the entire state space module from each Mamba layer, leaving only linear projections and convolution components; while effective for model size reduction, this approach can severely impact sequence modeling performance. In parallel, Ghattas et al. [9]

| Model | Garak Score | AEGIS Score |
|---|---|---|
| Nemotron-H-8B | 70.75 | 99.83 |
| Nemotron-H-4B | 67.77 | 98.17 |

Table 5: Safety scores before and after compression.

introduce a more granular pruning strategy, targeting state space dimension, Mamba head dimension, and head merging. Unfortunately, this approach has limited scalability, especially for large models requiring tensor parallelism; the proposed head merging also risks per-head information loss.

## 5 Conclusions

This paper has presented a novel group-aware pruning method for Mamba layers along with a unified *hybrid pruning recipe* that effectively combines Mamba pruning with FFN, embedding dimension and layer pruning. We have applied this unified recipe to the Nemotron-H 8B model to obtain NEMOTRON-H 4B, a compressed hybrid language model that achieves state-of-the-art accuracy and efficiency. Despite a 50% reduction in model size, NEMOTRON-H 4B retains over 96% of the original 8B model's accuracy, while requiring up to 40× fewer training tokens. Notably, our compression method also maintains the model's safety performance. NEMOTRON-H 4B advances the accuracy-efficiency Pareto frontier, achieving ∼2× faster inference and 2.6% higher accuracy across a diverse set of tasks.

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

# A  Technical Appendices and Supplementary Material

## A.1  FLAP Importance for Hybrid Models

FLAP [2] is a retraining-free structured pruning technique designed to measure the recoverability of a model's output feature map upon removing specific columns from weight matrices. FLAP quantifies the "fluctuation" of each input feature relative to a baseline using calibration data. Specifically, the FLAP importance score for a column is computed as the product of the squared norm of the column weights and the sample variance of the corresponding input features across calibration samples.

We extend FLAP to the SSM layers in hybrid architectures by applying the metric to the activations serving as inputs to the output projection ($W_O$). Here, we compute the FLAP importance by assessing the variance in activations input to the $W_O$, weighted by the squared norms of the respective columns of the $W_O$. Mathematically, the extended FLAP importance metric for a given column $j$ of weight matrix $W$ in SSM layers can be defined as:

$$S_j = \|W_j\|^2 \cdot \mathrm{Var}(X_j)$$

where $\|W_j\|^2$ denotes the squared norm of the column weights and $\mathrm{Var}(X_j)$ represents the variance of the activations input to the output projection matrix of SSM layer across calibration samples.

We use the above-computed metric to rank different heads within each group and remove the corresponding rows in the input projection matrix, the corresponding channels in the SSM convolution kernel, corresponding rows in the $A$ and $D$ matrices of SSM, as well as trimming the corresponding columns in the output projection matrix.

**Results.**  Table 6 shows that FLAP-based importance estimation yields mixed results before lightweight KD across pruning strategies. After KD, it performs on par with the L2-based approach when applied to candidate #1; it doesn't seem to offer any clear advantage, however.

| Pruning Type | Configuration | L2 LM Loss | FLAP LM Loss |
|---|---|---|---|
| Baseline | No pruning | 1.168 | 1.168 |
| FFN | FFN = 16384 | 1.364 | **1.32** |
|  | FFN = 11568 | 1.803 | **1.64** |
|  | FFN = 8192 | 2.281 | **1.95** |
| Attention | ATT Heads = 16 | **1.282** | 1.40 |
| Mamba (SSM) | Mamba Heads = 112 | **1.305** | 1.73 |
|  | Mamba Heads = 96 | **2.150** | 4.59 |
|  | Mamba Heads = 64 | **9.040** | 11.21 |
| Mixed | #1 | **3.690** | 5.854 |
|  | #1 + Lightweight KD | **1.380** | **1.380** |

Table 6: LM loss comparison when pruning different model components using L2 and FLAP metrics. Baseline: 128 Mamba heads, 21,504 FFN size, 32 attention heads.

## A.2  Depth Pruning

Figure 4 shows the average importance scores for each layer in the Nemotron-H 8B Base model, with green, blue, and red dotted lines representing self-attention, FFN, and Mamba layers. As seen in previous work [5], the most important layers are concentrated at the model's start and end. Interestingly, the first attention layer is among the least important, while other attention layers are more critical than neighboring layers. A "saw-like" pattern emerges where MLP layers are more important than adjacent Mamba layers in the middle of the network, though this reverses in the model's critical regions.

We experimented by pruning the least important layers (4, 8, 12, 16, and 26 layers), followed by distillation with 126B tokens. While core-knowledge benchmarks remained largely unaffected, tasks like math and coding showed significant performance degradation (Figure 5).

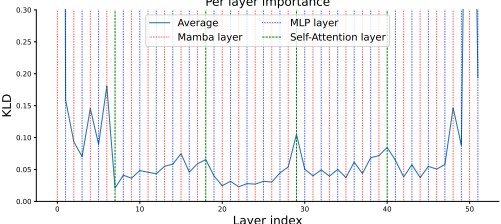

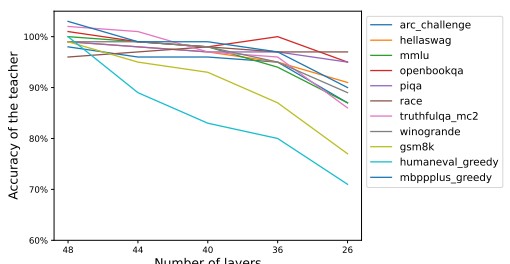

Figure 4: Layer importance measured as the KLD between logits of the full model and a model with that layer removed, averaged over a small training subset. Vertical dotted lines indicate layer types: self-attention (green), FFN (blue), and Mamba2 (red).

Figure 5: Accuracy drop relative to the 8B model across progressively depth-only pruned variants (48, 44, 40, 36, and 26 layers). Each model is directly pruned from the 8B and distilled using 126B tokens.

## A.3 Mamba Scoring

| FFN | Embedding Dim | Mamba | | LM-Loss | | |
| --- | --- | --- | --- | --- | --- | --- |
| | | Heads | Head Channels | $W_x$ | $W_z$ | $W_O$ |
| 12,288 | 3,072 | 112 | 64 | **3.56** | 4.11 | 3.79 |
| 13,056 | 3,072 | 112 | 56 | **3.59** | 6.61 | 5.30 |
| 13,056 | 3,072 | 96 | 64 | **4.49** | 5.39 | 4.49 |
| 14,592 | 3,072 | 96 | 56 | **4.68** | 7.09 | 10.01 |
| 12,288 | 3,072 | 128 | 56 | 5.98 | 5.43 | **4.99** |
| 13,824 | 3,072 | 128 | 48 | **5.99** | 6.01 | 9.47 |

Table 7: Mamba scoring ablation. The zero-shot LM-loss for top 6 pruned models based on Mamba scores calculated from activations of $W_x$, $W_z$, and $W_O$. The $W_x$ activations result in the best zero-shot LM-loss in most of the cases.

## A.4 Throughput and Latency Comparisons

Figure 6 compares the throughput and latency of NEMOTRON-H 4B to Phi-4 Mini 4B, Qwen-2.5 3B and Nemotron-H 8B.

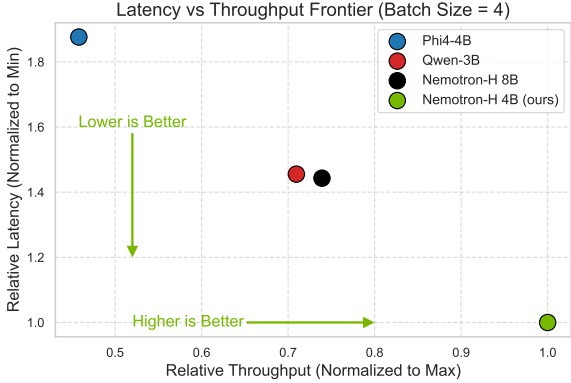

Figure 6: Throughput and latency comparisons across four models: Phi-4-Mini-4B, Qwen-2.5-3B, Nemotron-H 8B, and NEMOTRON-H 4B (ours). Relative throughput and latency represents are measured for an input and output context length of 65536 and 1024, respectively.

## A.5 Model Parameter Correlations

This section details the relationships between different model components for our 4B variants, illustrating the hyperparameter trade-offs required to maintain a fixed parameter budget (Figure 7).

Furthermore, we present an ablation study justifying our choice to optimize Mamba heads ($m_h$) over head channels ($m_d$), as $m_h$ pruning yields superior results across all metrics (Figure 8).

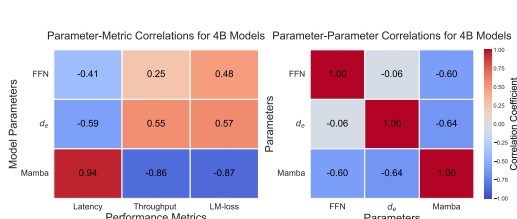

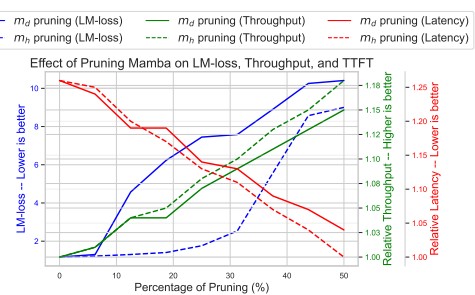

Figure 7: **Left:** Correlation matrix showing relationships between performance metrics and model components—FFN, embedding dimension ($d_e$), and Mamba parameters (varying both heads $m_h$ and head dimension $m_d$)—across 125 4B variants with fixed depth (52 layers). **Right:** Model parameter correlations for a fixed 4B parameter budget—highlighting trade-offs where increasing one component reduces others.

Figure 8: Impact of pruning Mamba heads ($m_h$) versus Mamba head channels ($m_d$) in isolation, with the rest of the network unchanged. Pruning $m_h$ consistently outperforms $m_d$ pruning across LM loss, latency, and throughput—establishing it as the preferred target for optimization.

## A.6 Generalizability to Mamba2

To evaluate the generalizability of our compression strategy to other models, we apply it to the Mamba2 1.3B model [7]. We prune the model to 780M parameters via SSM and embedding pruning, and then subsequently train the pruned model on 10.5B tokens. We compare our pruned 780M model to the Mamba2 780M and 1.3B models trained from scratch on 300B tokens [7].

As shown in Table 8, our compressed 780M model, despite being trained on significantly fewer tokens (10.5B vs. 300B), outperforms the 780M model trained from scratch and achieves an average score comparable to the original 1.3B model. These results provide further insights into the generalizability of our compression method.

Table 8: Comparison of our compressed Mamba2 780M model against Mamba2 780M and 1.3B models trained from scratch [7]. Despite being trained on significantly fewer tokens (10.5B vs. 300B), our compressed model achieves a better average score than the 780M baseline.

| Benchmark | Mamba2 780M | Mamba2 1.3B | Compressed 780M |
|---|---|---|---|
| arc_challenge | 28.6 | 33.2 | 34.2 |
| arc_easy | 54.7 | 60.6 | 60.4 |
| commonsense_qa | 19.6 | 20.9 | 26.4 |
| hellaswag | 54.7 | 59.9 | 50.4 |
| openbookqa | 36.4 | 37.0 | 34.2 |
| piqa | 72.1 | 73.5 | 71.3 |
| race | 21.8 | 24.8 | 32.5 |
| social_iqa | 41.0 | 42.9 | 41.1 |
| truthfulqa_mc2 | 38.1 | 36.1 | 38.9 |
| winogrande | 58.0 | 60.1 | 57.7 |
| **Avg** | **42.5** | **44.9** | **44.7** |
| Tokens | 300B | 300B | 10.5B |

## A.7 Compute Resources

All experiments were performed on 16-32 × NVIDIA DGX H100 nodes (8× H100 80GB) for short turnaround times.

