# OpenReview forum: "Efficient Hybrid Language Model Compression through Group-Aware SSM Pruning"
_NeurIPS.cc/2025/Conference — NeurIPS 2025 poster_

### Official Review · Reviewer_tXTL · 2025-06-09

**Clarity:** 3
**Significance:** 3
**Originality:** 2
**Rating:** 5
**Confidence:** 4

**Summary:**

This paper explores width, depth, and head pruning for hybrid Mamba architectures. In particular, it proposes a novel method for pruning SSM (State Space Model) modules.

**Questions:**

I am confused about the **Group-Aware Head Permutation Constraints** and **Head Channel Consistency**. Prior work [1] suggests that the channel dimension in the Mamba architecture exhibits **permutation invariance**. Why, then, is it necessary to maintain **uniform pruning across all heads within a group**, rather than allowing permutation of heads across groups?

[1] Hung-Yueh Chiang et al.,  Quamba2: A Robust and Scalable Post-training Quantization Framework for Selective State Space Models

**Ethical Concerns:**

["NO or VERY MINOR ethics concerns only"]

**Final Justification:**

Thank you for your response. I believe your explanation has effectively resolved my confusion.

**Limitations:**

The authors transparently acknowledge all of the limitations I was able to identify within the paper itself.

**Paper Formatting Concerns:**

No major formatting issues observed.

**Quality:**

3

**Strengths And Weaknesses:**

### **Strengths**:

- This is the first work, to the best of my knowledge, that conducts an in-depth analysis of different pruning strategies for the Mamba architecture. The pruning strategy is systematically explored across multiple architectural axes (depth, width, and heads).



### **Weaknesses**:
- The paper lacks sufficient **discussion of related work** and **stronger baseline methods**, such as [2]. Including these would improve the completeness of the evaluation.
- In addition, the **experimental scope is limited**—the evaluation is conducted solely on **Nemotron-H-8B-Base**, which makes it difficult to assess the general applicability of the proposed method. Broader experiments on additional models are recommended.

---

> ### Author Rebuttal · Authors · 2025-07-31
>
> We thank the reviewer for their valuable insights and comments. We have tried our best to provide the responses for the questions as follows:
> # Weaknesses:
>
> ## Reviewer: discussion of related work and stronger baseline methods
> At submission, we were aware of only two Mamba pruning papers (refs. [7] and [16]), both cited. For transformer pruning (self-attention and FFN), we referenced [4], [17], and [20]. Regarding ref. [2] mentioned by the reviewer, assuming they’re referring to ref. [2] in the paper, we extended the FLAP methodology to Mamba in Appendix A1 and included results. We're happy to add any missing references if advised by the reviewer.
>
> ## Reviewer: the evaluation is conducted solely on Nemotron-H-8B-Base
>
> To help answer the question on whether our method generalizes to other hybrid models, we have applied our compression strategy to the original Mamba2 1.3B model [6], trimmed it down to 780M parameters via ssm and embedding pruning, and then further trained the pruned model on 10.5B tokens. We compare our pruned 780M model to the Mamba2 780M model trained from scratch on 300B tokens [6]. Given the short time limit of the rebuttal and limited compute, this was the largest model we could evaluate; however, we hope that these Mamba2 pruning results provide further insights into the generalizability of our method.
> As shown in the table below, our compressed 780M model outperforms the model trained from scratch.
>
> | Benchmark        | Mamba2 780M | Mamba2 1.3B | Compressed 780M |
> |------------------|-------------|-------------|------------------|
> | arc_challenge    | 28.6        | 33.2        | 34.2             |
> | arc_easy         | 54.7        | 60.6        | 60.4             |
> | commonsense_qa   | 19.6        | 20.9        | 26.4             |
> | hellaswag        | 54.7        | 59.9        | 50.4             |
> | openbookqa       | 36.4        | 37.0        | 34.2             |
> | piqa             | 72.1        | 73.5        | 71.3             |
> | race             | 21.8        | 24.8        | 32.5             |
> | social_iqa       | 41.0        | 42.9        | 41.1             |
> | truthfulqa_mc2   | 38.1        | 36.1        | 38.9             |
> | winogrande       | 58.0        | 60.1        | 57.7             |
> | **Avg**          | **42.5**    | **44.9**    | **44.7**         |
> | **Tokens**       | **300B**    | **300B**    | **10.5B**        |
>
> We will include these results in the supplementary materials.
> [6] Tri Dao and Albert Gu. Transformers are SSMs: Generalized models and efficient algorithms through 320 structured state space duality. arXiv preprint arXiv:2405.21060, 2024.
>
> # Questions
> We thank the reviewer for providing the related reference, and we will make sure to cite it in our manuscript under related works since it is directly related to making SSMs and hybrid models more efficient.
> ## Reviewer: Group-Aware Head Permutation Constraints and Head Channel Consistency
> As described in the paper, there are two constraints, both of which are directly due to the triton kernels implemented for mamba from https://github.com/state-spaces/mamba, which we use in our implementation.:
>
> Constraint 1: the constraints raised by the same channel permutation for all heads could be observed in multiple instances, a place where it is easy to follow is during the decoding of Mamba2. During token decoding, the latent SSM state, h, whose dimension is mamba head times mamba channels times SSM state dimension (m_h x m_d x d_s), is updated by Bx (equation 3 in the manuscript), and therefore, all the head channels in “x” should follow the same channel indexing in “h”.
>
> We note that this constraint could be relaxed if one modifies all the operations to handle different rankings, but this needs further changes to the Triton kernel, and we leave it for future studies. To this point, while [1] cited by the reviewer is correct, it does not account for the implementation details of the Mamba kernels.
>
> At the end of our response, we have also provided a simple snippet simulating a decoding step in SSMs, and showcase why inconsistent head permutation across heads can result in different outputs mathematically.
>
> Constraint 2: The constraints raised by groups that apply to the mamba heads, as shown in Figure 3 in the manuscript, is due to broadcasting of the SSM matrix “B”, whose dimension is the number of SSM groups times SSM state dimension (g x d_s), which is handled by replicating the d_s parameters for each group for the whole mamba heads assigned to that group in the input “x” matrix. Therefore, to allow heads to be permuted across different groups, one would need to implement a new Triton kernel to allow different broadcasting mechanisms. We leave this for future studies.
>
> We would also like to note that, in many of the previous hybrid models, such as the one considered in reference [7], there is only one mamba group, so in that case, mamba heads can be permuted freely.
>
> ```python #
> import torch
> from einops import rearrange, repeat
> import torch.nn.functional as F
> import os
> import random
> import numpy as np
>
> def set_all_seeds(seed):
>     random.seed(seed)
>     os.environ['PYTHONHASHSEED'] = str(seed)
>     np.random.seed(seed)
>     torch.manual_seed(seed)
>     torch.cuda.manual_seed(seed)
>     torch.backends.cudnn.deterministic = True
>     torch.backends.cudnn.benchmark = False
>
> seed = 42
> set_all_seeds(seed)
>
> #Toy example parameters
> d_state = 5
> d_inner_local=4*2
> ngroups_local=2
> headdim=4
> nheads=d_inner_local//headdim
>
> ssm_state=torch.rand(nheads, headdim, d_state)
> x = torch.rand(d_inner_local)
>
> B = torch.rand(d_state*ngroups_local)
> C = torch.rand(d_state*ngroups_local)
> dt = torch.rand(nheads)
> dt_bias = torch.rand(nheads)
> A = torch.rand(nheads)
> D = torch.rand(nheads)
>
> B = rearrange(B, "(g n) -> g n", n=d_state)
> C = rearrange(C, "(g n) -> g n", n=d_state)
> B = repeat(B, "g n -> (g h) n", h=d_inner_local // ngroups_local)
> C = repeat(C, "g n -> (g h) n", h=d_inner_local // ngroups_local)
>
> dt = repeat(dt, "h -> (h p)", p=headdim)
> dt_bias = repeat(dt_bias, "h -> (h p)", p=headdim)
> A = repeat(A, "h -> (h p) n", p=headdim, n=d_state)
> D = repeat(D, "h -> (h p)", p=headdim)
>
>
> def process(dt,dt_bias, A, D, x, ssm_state, B, C):
>     dt = F.softplus(dt + dt_bias.to(dtype=dt.dtype))
>     dA = torch.exp(torch.einsum("d,dn->dn", dt, A))
>
>     dB_x = torch.einsum('d,dn,d->dn', dt, B, x)
>     ssm_state.copy_(
>         ssm_state * rearrange(dA, "(h p) n -> h p n", p=headdim)
>         + rearrange(dB_x, "(h p) n -> h p n", p=headdim)
>     )
>
>     y = torch.einsum(
>         "dn,dn->d",
>         rearrange(ssm_state, "h p n -> (h p) n", p=headdim),
>         C,
>     )
>     y = y + D * x
>     return y
>
>
>
> channel_permutation = [1,0,3,2]
>
> # Change channel_permutation_for_two_heads to [1, 0, 3, 2, 4, 5, 6, 7] for inconsistent channel permutation to observe the differntt output
> channel_permutation_for_two_heads = [1, 0, 3, 2, 5, 4, 7, 6]  # uncomment for CORRECT
> #channel_permutation_for_two_heads = [1, 0, 3, 2, 4, 5, 6, 7]  # uncomment for INCORRECT
>
> x_permuted = x[channel_permutation_for_two_heads]
> ssm_state_permuted_channel = ssm_state[:,channel_permutation,:]
>
> y = process(dt,dt_bias, A, D, x, ssm_state, B, C)
> y_permuted = process(dt, dt_bias, A, D, x_permuted, ssm_state_permuted_channel, B, C)
> print(abs(y_permuted[channel_permutation_for_two_heads]-y).sum()) # should be zero for correct and non-zero for incorrect permutation
>
> ```

---

> > ### Comment · Reviewer_tXTL · 2025-08-04
> >
> > Thank you for your response.
> >
> > First of all, I would like to apologize for my oversight — I forgot to include a proper citation to FLAP.
> >
> > That said, I believe your explanation has effectively resolved my confusion.

---

### Official Review · Reviewer_WxVs · 2025-07-03

**Clarity:** 3
**Significance:** 3
**Originality:** 2
**Rating:** 5
**Confidence:** 3

**Summary:**

This paper proposes a group-aware pruning method for the Mamba layer. It combines Mamba pruning with pruning of other network layers to produce smaller and more accurate models while reducing training costs.
This paper uses this method to compress the parameters of the Nemotron-H 8B hybrid model, reducing training tags, increasing throughput, and improving accuracy compared to models with the same parameters.

**Questions:**

1. Can the pruning method show similar effects on models other than Nemotron-H 8B?
2. Comparison of the compressed model with the model of the same series with smaller parameters.
3. Experiments and analysis of inference throughput.

**Ethical Concerns:**

["NO or VERY MINOR ethics concerns only"]

**Final Justification:**

Many of my concerns were addressed by the authors, and I encourage them to include new results and interpretations in the final paper.

Thank you for your additional analysis, we would like to raise our rating to 5

**Limitations:**

yes

**Quality:**

3

**Strengths And Weaknesses:**

**Strengths**

This paper focuses on pruning and distillation, and the proposed pruning method is intuitively reasonable. A relatively detailed experiment and analysis are conducted, and 7 findings are proposed and corresponding experimental results are provided. The compressed model of Nemotron-H 8B also achieves improvements over the same parameter model in knowledge, mathematics, encoding, common sense reasoning, and reading comprehension tasks.

**Weaknesses**

This paper only distills Nemotron-H 8B, which makes people wonder whether the pruning scheme can be extended. I hope the authors can provide more than one compressed model and show similar results. Does the Mamba layer limit the range of LLMs that can be pruned?
In the main experiment, the Nemotron-H-4B compressed model is only used for comparison with the non-Nemotron series baseline. In order to eliminate the interference of model quality, can you provide the comparison results of the compressed model with the model with smaller parameters in the same series? For example, compress Qwen-7B to 3B and compare the compressed model with Qwen-3B?
The author mentioned that the compressed model achieved 2 times faster inference throughput, and experiments and analysis of this conclusion are needed.

---

> ### Author Rebuttal · Authors · 2025-07-31
>
> We thank the reviewer for their valuable insights and comments. We have tried our best to provide the responses for the questions as follows:
> # Weaknesses
> ## Reviewer: Can the pruning method show similar effects on models other than Nemotron-H 8B?
> To help answer the question on whether our method generalizes to other hybrid models, we have applied our compression strategy to the original Mamba2 1.3B model [6], pruned it down to 780M parameters via ssm and embedding pruning, and then further trained the pruned model on 10.5B tokens. We compare our pruned 780M model to the Mamba2 780M model trained from scratch on 300B tokens [6]. Given the short time limit of the rebuttal and limited compute, this was the largest model we could evaluate; however, we hope that these Mamba2 pruning results provide further insights into the generalizability of our method.
> As shown in the table below, our compressed 780M model outperforms the model trained from scratch.
>
> | Benchmark        | Mamba2 780M | Mamba2 1.3B | Compressed 780M |
> |------------------|-------------|-------------|------------------|
> | arc_challenge    | 28.6        | 33.2        | 34.2             |
> | arc_easy         | 54.7        | 60.6        | 60.4             |
> | commonsense_qa   | 19.6        | 20.9        | 26.4             |
> | hellaswag        | 54.7        | 59.9        | 50.4             |
> | openbookqa       | 36.4        | 37.0        | 34.2             |
> | piqa             | 72.1        | 73.5        | 71.3             |
> | race             | 21.8        | 24.8        | 32.5             |
> | social_iqa       | 41.0        | 42.9        | 41.1             |
> | truthfulqa_mc2   | 38.1        | 36.1        | 38.9             |
> | winogrande       | 58.0        | 60.1        | 57.7             |
> | **Avg**          | **42.5**    | **44.9**    | **44.7**         |
> | **Tokens**       | **300B**    | **300B**    | **10.5B**        |
>
> We will include these results in the supplementary materials.
> [6] Tri Dao and Albert Gu. Transformers are SSMs: Generalized models and efficient algorithms through 320 structured state space duality. arXiv preprint arXiv:2405.21060, 2024.
>
> ## Reviewer: Does the Mamba layer limit the range of LLMs that can be pruned?
> Adding Mamba layer pruning support expands the range of LLMs that can be compressed by providing more axes to prune the model along. Prior works [20] were limited to compression of LLMs with self attention and FFN layers only. Our addition of Mamba layer pruning extends this to support hybrid models (with Mamba layers) and pure Mamba models as well.
>
> ## Reviewer: Comparison of the compressed model with the model of the same series with smaller parameters.
> While no 4B hybrid model with the same architecture trained from scratch exists, we address this concern with additional results on compressing Mamba 1.3B → 780M, included in the response to the weaknesses above.
>
> ## Reviewer: Experiments and analysis of inference throughput
> Inference and throughput were measured using the Megatron-LM repository (https://github.com/NVIDIA/Megatron-LM/tree/main/megatron ) for all the models. Figure 6 shows a relative 2x speedup for the compressed 4B model over the previous SOTA, Phi-4-Mini-4B. We additionally provide the table below with absolute numbers for the same setting (64k input, 1k output).
>
> | Model                 | Throughput (tokens/s/GPU) | Latency (time to first token/s) |
> |-----------------------|----------------------------|----------------------------------|
> | Phi-4-Mini-4B         | 131.7                      | 12.384                           |
> | Qwen-2.5-3B           | 204.0                      | 9.606                            |
> | Nemotron-H 8B         | 212.5                      | 9.524                            |
> | COMPRESSED-HYBRID 4B  | 287.5                      | 6.6                              |
>
> We also note that the data on how different pruning axes affect latency and throughput are available in Figure 7 and 8 in the manuscript, and the relative inference throughput numbers for the candidates we provide in Table 1 is available in the last column of the same table.
>
> #Questions
>
> ## Reviewer: Can the pruning method show similar effects on models other than Nemotron-H 8B? Comparison of the compressed model with the model of the same series with smaller parameters.
>
> Please refer to our response above on the added study on compressing Mamba2 1.3b to 780m.
>
> ## Reviewer: Experiments and analysis of inference throughput.
>
> Please refer to our response in the final section addressing the identified weaknesses.

---

> > ### Comment · Reviewer_WxVs · 2025-08-07
> >
> > Many of my concerns were addressed by the authors, and I encourage them to include new results and interpretations in the final paper.
> >
> > Thank you for your additional analysis, we would like to raise our rating to 5

---

### Official Review · Reviewer_D6BM · 2025-07-03

**Clarity:** 3
**Significance:** 2
**Originality:** 3
**Rating:** 5
**Confidence:** 4

**Summary:**

This paper introduces a ​​compression framework​ for hybrid Transformer-State Space Models, the innovations include:
- A group-aware pruning method for Mamba layers that preserves SSM block structure and sequence modeling capabilities.
- A unified hybrid pruning recipe

The proposed method compresses the Nemotron-H 8B model to 4B parameters with much less training tokens and comparable performances.

**Questions:**

- How knowledge distillation affects the performances? Results with pruning only before distillation can be ablated to showcase the effectiveness of proposed pruning techniques.
- I wonder the generalization ability of this method, what will the performance be if the proposed method is applied on other open-sourced models such as jamba.

**Ethical Concerns:**

["NO or VERY MINOR ethics concerns only"]

**Final Justification:**

The paper proposes a meaningful for mamba-based model pruning. And my two questions are basically addressed by the rebuttal.

**Limitations:**

Yes

**Quality:**

3

**Strengths And Weaknesses:**

strength
- The paper is well written and well-motivated.
- The 7 findings provide meaningful and comprehensive guidance for SSM-based model compression in the community.
- The performance and efficiency of the final model is well evaluated and achieves strong performances.

---

> ### Author Rebuttal · Authors · 2025-07-31
>
> We thank the reviewer for their valuable insights and comments. We have tried our best to provide the responses for the questions as follows:
>
> # Questions
>
> ## Reviewer: How knowledge distillation affects the performances? Results with pruning only before distillation can be ablated to showcase the effectiveness of proposed pruning techniques.
>
> Thank you for this point. Assuming performance here refers to benchmark scores, we note that the pruned model initially performs near-randomly, but retains core capacity to recover through distillation. In terms of LM loss, it starts at 3.3 and improves to 1.15 after full knowledge distillation.
>
> We analyze the impact of pruning along different axes on LM loss (without distillation) and speed in Figures 7 and 8. As noted in Section 3.1, we prune 125 candidates and select the best 22 based on LM loss post-pruning. However, our main selection metric across different pruning techniques/candidates is LM loss after short (6B token) distillation, as shown in Table 1, since better LM loss post-pruning does not always translate to better post-distillation performance. This behavior has been demonstrated in [17]. If advised, we will add the above details to the manuscript.
>
> [17] Saurav Muralidharan, Sharath Turuvekere Sreenivas, Raviraj Joshi, Marcin Chochowski, Mostofa Patwary, Mohammad Shoeybi, Bryan Catanzaro, Jan Kautz, and Pavlo Molchanov. Compact language models via pruning and knowledge distillation. arXiv preprint arXiv:2407.14679, 2024.
>
>
> ## Reviewer: I wonder the generalization ability of this method, what will the performance be if the proposed method is applied on other open-sourced models such as jamba.
>
> To help answer the question on whether our method generalizes to other Mamba based models, we have applied our compression strategy to the original Mamba2 1.3B model [6], pruned it down to 780M parameters via ssm and embedding pruning, and then further trained the pruned model on 10.5B tokens. We compare our pruned 780M model to the Mamba2 780M model trained from scratch on 300B tokens [6]. Unfortunately, due to the very short response period, we couldn’t apply our method to Jamba; however, we hope that these Mamba2 pruning results provide further insights into the generalizability of our method.
>
> As shown in the table below, our compressed 780M model outperforms the model trained from scratch.
>
> | Benchmark        | Mamba2 780M | Mamba2 1.3B | Compressed 780M |
> |------------------|-------------|-------------|------------------|
> | arc_challenge    | 28.6        | 33.2        | 34.2             |
> | arc_easy         | 54.7        | 60.6        | 60.4             |
> | commonsense_qa   | 19.6        | 20.9        | 26.4             |
> | hellaswag        | 54.7        | 59.9        | 50.4             |
> | openbookqa       | 36.4        | 37.0        | 34.2             |
> | piqa             | 72.1        | 73.5        | 71.3             |
> | race             | 21.8        | 24.8        | 32.5             |
> | social_iqa       | 41.0        | 42.9        | 41.1             |
> | truthfulqa_mc2   | 38.1        | 36.1        | 38.9             |
> | winogrande       | 58.0        | 60.1        | 57.7             |
> | **Avg**          | **42.5**    | **44.9**    | **44.7**         |
> | **Tokens**       | **300B**    | **300B**    | **10.5B**        |
>
> We will include these results in the supplementary materials.
>
> [6] Tri Dao and Albert Gu. Transformers are SSMs: Generalized models and efficient algorithms through 320 structured state space duality. arXiv preprint arXiv:2405.21060, 2024.

---

### Official Review · Reviewer_2zWJ · 2025-07-23

**Clarity:** 3
**Significance:** 3
**Originality:** 4
**Rating:** 5
**Confidence:** 4

**Summary:**

The paper proposes a group-aware pruning method for the Mamba layers, which preserves the structural integrity of SSM blocks and their sequence modeling capabilities. The authors combine this with embedding, FFN, and layer pruning to further reduce the parameter count, and they conduct extensive ablations to study the effect of pruning different components on accuracy, latency, and throughput — while keeping the total parameter count fixed at 4B. The proposed pruning method enables compression model by 50% with minimal drop in performance.

**Questions:**

1. How well does the proposed pruning strategy generalize for different family architectures?

2. How does the pruning affect the model bias for hybrid LLMs?

**Ethical Concerns:**

["NO or VERY MINOR ethics concerns only"]

**Final Justification:**

The authors have addressed most of the concerns and have included new experiments with different model families and analysis on model bias.

**Limitations:**

yes

**Quality:**

4

**Strengths And Weaknesses:**

### Strengths
1. Hybrid attention is critical for long context performance; however, pruning hybrid architecture is not well studied. This paper presents a compression strategy that combines SSM pruning with the pruning of other network components such as FFN neurons, embedding channels, and network depth. Authors also conducted an in-depth architectural search for pruning different components (different layers, depth vs width, etc).

2. As demonstrated in Figure 3, the proposed group-aware pruning method for the Mamba layer preserves the structural representation of the Mamba blocks and provides a simple method for pruning hybrid LLMs.

3.  Extensive ablations are conducted in Section 3.1  to compare different pruning strategies and the impact of pruning different components on validation loss, latency and throughput (Table 1).

4. As shown in Table 2, the proposed pruning strategy achieves 50% compression rate while retaining  over 96% of the original 8B model’s accuracy, while improving throughput by ∼1.4x – demonstrating the efficacy of the method.

5. Pushes Pareto-frontier for throughput, though results are limited to the Nemotron family.

### Weaknesses

1. The major limitation of this work (as the authors also noted) is that all the experiments are conducted with the Nematron model; it is not clear if the observation and analysis will hold true for different model families.

2. The model performance deteriorates after pruning, which is later recovered using distillation.

3. There is no ablation on pruning heuristic used in sec 2.2.

4. No discussion on the potential impact of pruning strategy proposed on the model bias, as prior work has highlighted that model compression exacerbates model bias.[1].

[1]. Gonçalves and Strubell, Understanding the Effect of Model Compression on Social Bias in Large Language Models

---

> ### Author Rebuttal · Authors · 2025-07-31
>
> We thank the reviewer for their insightful questions and discussions. We will try our best to address the questions and concerns below:
>
> # Weaknesses
> ## Reviewer: it is not clear if the observation and analysis will hold true for different model families
>
> To help answer the question on whether our method generalizes to other hybrid models, we have applied our compression strategy to the original Mamba2 1.3B model [6], trimmed it down to 780M parameters via ssm and embedding pruning, and then further trained the pruned model on 10.5B tokens. We compare our pruned 780M model to the Mamba2 780M model trained from scratch on 300B tokens [6]. Given the short time limit of the rebuttal and limited compute, this was the largest model we could evaluate; however, we hope that these Mamba2 pruning results provide further insights into the generalizability of our method.
> As shown in the table below, our compressed 780M model outperforms the model trained from scratch.
>
> | Benchmark        | Mamba2 780M | Mamba2 1.3B | Compressed 780M |
> |------------------|-------------|-------------|------------------|
> | arc_challenge    | 28.6        | 33.2        | 34.2             |
> | arc_easy         | 54.7        | 60.6        | 60.4             |
> | commonsense_qa   | 19.6        | 20.9        | 26.4             |
> | hellaswag        | 54.7        | 59.9        | 50.4             |
> | openbookqa       | 36.4        | 37.0        | 34.2             |
> | piqa             | 72.1        | 73.5        | 71.3             |
> | race             | 21.8        | 24.8        | 32.5             |
> | social_iqa       | 41.0        | 42.9        | 41.1             |
> | truthfulqa_mc2   | 38.1        | 36.1        | 38.9             |
> | winogrande       | 58.0        | 60.1        | 57.7             |
> | **Avg**          | **42.5**    | **44.9**    | **44.7**         |
> | **Tokens**       | **300B**    | **300B**    | **10.5B**        |
>
> We will include these results in the supplementary materials.
> [6] Tri Dao and Albert Gu. Transformers are SSMs: Generalized models and efficient algorithms through 320 structured state space duality. arXiv preprint arXiv:2405.21060, 2024.
>
> ## Reviewer: model performance deteriorates after pruning
> The considerable deterioration in model performance right after pruning, and before applying knowledge distillation, is expected and universal (e.g. [17]; this is because some of the heads, channels, embedding dimension, and MLPs that contribute to the final logits are removed from the model. However, this degradation, although major, is recoverable to a significant degree with minimal data and knowledge distillation, as shown in the paper.
>
> ## Reviewer: ablation on pruning heuristic used in sec 2.2
> We would like to point out that since the aggregation metric for FFN and Layernorms is not Mamba-dependent, we leverage the ablations from [17]. However, if the reviewer advises so, we would be happy to add this ablation to the appendix. Regarding the depth pruning and FLAP importance in Section 2.2, the ablations and heuristics are available in Section A.2 and A.1 of the appendix provided with the submitted manuscript.
>
> ## Reviewer: potential impact of pruning strategy proposed on the model bias
> Thank you for highlighting this important concern. We evaluated Garak and AEGIS scores for the compressed 4B model and its 8B parent, observing similar relative differences as in the evaluation benchmarks. This indicates the safety scores are as expected for the compressed model. We will include these results and the discussion in the supplementary materials.
> | Model              | Garak Score | AEGIS Score |
> |--------------------|-------------|-------------|
> | Nemotron-H-8B      | 70.75%      | 99.83%      |
> | Compressed-4B      | 67.77%      | 98.17%      |
>
> # Questions
>
>
> ## Reviewer: How well does the proposed pruning strategy generalize for different family architectures?
>
> Please refer to the Mamba2 1.3b to 780m compression discussed above
>
> ## Reviewer: how pruning affects the model bias
>
> Please refer to our answer on model safety and bias above.

---

### Decision · Program_Chairs · 2025-09-17

**Decision:**

Accept (poster)

**Comment:**

The paper seems to be one of the first in-depth studies on pruning hybrid SSM architectures, focusing on Mamba layers. The empirical results are quite strong: the authors demonstrate a 50% compression of the Nemotron-H 8B model down to 4B parameters, retaining most of the original model's accuracy. The reviewers noted extensive ablations presented in the paper.

The most common concern was that the experiments were performed only on a single model, Nemotron-H 8B, making it difficult to assess if the method generalizes to other model families. The authors added new experiments during the rebuttal, which the reviewers found convincing. Another concern was model bias post-compression. Once again, additional experiments during the rebuttal were added on some safety benchmarks. Some clarity issues were also noted, which will hopefully be fixed in the camera-ready based on the provided feedback.